# IL-10 producing type 2 innate lymphoid cells prolong islet allograft survival

Qingsong Huang[1,†], Xiaoqian Ma[2,3,†], Yiping Wang[2], Zhiguo Niu[1], Ruifeng Wang[2], Fuyan Yang[4], Menglin Wu[1], Guining Liang[5], Pengfei Rong[3], Hui Wang[1], David CH Harris[2], Wei Wang[3,*] & Qi Cao[1,2,**] (ID)

## Abstract

Type 2 innate lymphoid cells (ILC2s) are a subset of ILCs with critical roles in immunoregulation. However, the possible role of ILC2s as immunotherapy against allograft rejection remains unclear. Here, we show that IL-33 significantly prolonged islet allograft survival. IL-33-treated mice had elevated numbers of ILC2s and regulatory T cells (Tregs). Depletion of Tregs partially abolished the protective effect of IL-33 on allograft survival, and additional ILC2 depletion in Treg-depleted DEREG mice completely abolished the protective effects of IL-33, indicating that ILC2s play critical roles in IL-33-mediated islet graft protection. Two subsets of ILC2s were identified in islet allografts of IL-33-treated mice: IL-10 producing ILC2s (ILC2[10]) and non-IL-10 producing ILC2s (non-ILC[10]). Intravenous transfer of ILC2[10] cells, but not non-ILC[10], prolonged islet allograft survival in an IL-10-dependent manner. Locally transferred ILC2[10] cells led to long-term islet graft survival, suggesting that ILC2[10] cells are required within the allograft for maximal suppressive effect and graft protection. This study has uncovered a major protective role of ILC2[10] in islet transplantation which could be potentiated as a therapeutic strategy.

**Keywords** IL-10; IL-33; innate lymphoid cells; islet transplantation; type 1 diabetes

**Subject Categories** Immunology; Metabolism; Stem Cells & Regenerative Medicine

## Introduction

Innate lymphoid cells (ILCs) are a novel group of immune cells with critical roles in immunity, tissue homeostasis, and pathological inflammation (Eberl *et al*, 2015; Sonnenberg & Artis, 2015; Vivier *et al*, 2018). ILCs are subdivided into three groups: ILC1, ILC2, and ILC3, based on their cytokine profiles and expression of specific transcription factors, mirroring the classification of CD4[+] T helper cell subsets into TH1, TH2, and TH17 cells. Type 2 ILCs (ILC2s) resemble TH2 cells as they require the transcription factor GATA-3; produce type 2 cytokines IL-4, IL-5, IL-9, and IL-13; and play important roles in immunity against pathogens and type 2 inflammation (Krabbendam *et al*, 2018). ILC2s also promote tissue recovery following acute injury in multiple organs, such as lung, intestine, and kidney (Monticelli *et al*, 2011; Cao *et al*, 2018). For example, in influenza virus infection of mice, ILC2s were activated by lung epithelial cell-derived IL-25 and IL-33 and promoted repair of the airway epithelium via producing amphiregulin (Areg) (Monticelli *et al*, 2011). More recently, IL-10 producing ILC2s, namely ILC2[10], have been identified in lung where they play important roles in resolution of lung inflammation (Seehus *et al*, 2017), indicating that ILC2s include different functional subsets. However, the possible immunoregulatory role of ILC2 in transplant rejection has not been addressed so far. Modulation of ILC2 activity may provide a therapeutic approach to maintain allograft tolerance.

Type 1 diabetes mellitus (T1DM) is an autoimmune disease in which pancreatic β cells are destroyed by autoreactive T cells, resulting in lifelong insulin dependency (Burrack *et al*, 2017; Paschou *et al*, 2018). Pancreatic islet transplantation is a promising treatment option for patients with type 1 diabetes that restores both endogenous insulin production and glycemic stability (Anazawa *et al*, 2019). However, islet graft rejection caused by immune cells remains one of the main obstacles to successful transplantation. Although advances in immunosuppressive therapies have promoted excellent short-term graft survival after islet transplantation,

1 Henan Key Laboratory of Immunology and Targeted Drugs, School of Laboratory Medicine, Xinxiang Medical University, Xinxiang, China
2 Centre for Transplant and Renal Research, Westmead Institute for Medical Research, The University of Sydney, Sydney, NSW, Australia
3 The Institute for Cell Transplantation and Gene Therapy, The Third Xiangya Hospital of Central South University, Changsha, China
4 The Department of Nephrology, First People's Hospital of Xinxiang Medical University, Xinxiang, China
5 The Department of Physiology, Guangxi Medical University, Nanning, China
*Corresponding author. Tel: +86 731 88618411; E-mail: cjr.wangwei@vip.163.com
**Corresponding author. Tel: +61 02 86273512; E-mail: qi.cao@sydney.edu.au
†These authors contributed equally to this work

immunosuppressive drugs with severe side effects remain ineffective at preventing late-stage allograft rejection (Gibly et al, 2011; Anazawa et al, 2019). Therefore, it is critical to develop applicable strategies that specifically target anti-islet immune responses to achieve long-term graft tolerance without use of immunosuppressive drugs. One attractive alternative therapy to prevent allograft rejection relies on harnessing the potential of regulatory T cells (Treg) (Gagliani et al, 2010; Lam et al, 2017). Recent studies have shown that ex vivo-expanded human Treg can prevent the development of islet and skin allograft rejection in a humanized mouse model (Issa et al, 2010; Yi et al, 2012). IL-33 significantly prolonged allograft survival in organ transplantation partially via increasing numbers of Tregs (Turnquist et al, 2011; Matta et al, 2016). We recently reported that IL-33-expanded kidney resident ILC2s prevented renal ischemia-reperfusion injury (IRI) via production of Areg (Cao et al, 2018). In the present study, we sought to determine the role of the IL-33-ILC2 pathway in islet allograft survival. Here, we report that IL-33 induced long-term survival of islet allografts via increasing both Tregs and ILC2s in vivo. Importantly, we further demonstrated that ex vivo-expanded ILC2s significantly prolonged allograft survival in an IL-10-dependent manner.

## Results

### IL-33 prevented rejection of islet allograft

To determine whether IL-33 could prolong islet graft survival, we treated diabetic C57BL/6 mice with mouse recombinant IL-33 (0.3 μg/mouse/day, intraperitoneally) for five consecutive days prior to islet transplantation (Fig 1A). As expected, blood glucose measurement showed that PBS-treated control mice all rejected their grafts rapidly, with a mean survival time of 12 days. In contrast, pretreatment with IL-33 led to prominent long-term graft survival (Fig 1B). In this case, a small proportion of the mice rejected their grafts around days 14–20, but the remaining 75 % ($n$ = 9 out of 12) of the mice retained their grafts indefinitely (> 80 days) (Fig 1C). Intraperitoneal glucose tolerance tests (IPGTTs) were performed to investigate islet graft function in vivo. The glucose tolerance in islet transplant mice treated with IL-33 was significantly improved compared with mice receiving islets alone (Fig 1D). The values of the area under the glucose curve (AUC) for IPGTT in the group with IL-33 treatment were significantly smaller than in those receiving islets alone (Fig 1E). Histology in PBS-treated control mice showed that islet architecture was lost, with only a few remaining insulin-

positive cells and massive leukocyte infiltration. In contrast, the group with IL-33 treatment showed well-preserved islet morphology with reduced/minimal leukocyte infiltration (Fig 1F). These results showed that IL-33 treatment prevented islet allograft rejection and improved islet function. We also found that IL-33 treatment in STZ-induced diabetic mice without islet transplantation improved fasting and non-fasting glycemia at day 15 and 18 post-STZ injection, but did not enhance survival of STZ-induced diabetic mice (Appendix Fig S1). The improved glycemia status of diabetic mice with IL-33 treatment could possibly contribute to prolonged islet allograft survival.

### IL-33 induced Th2 cytokine, ILC2s, and regulatory T cells in vivo

We then investigated the mechanism by which IL-33 prevents rejection of islet allografts. In islet transplant mice treated with IL-33, the serum levels of the Th1-related cytokines IFN-γ and IL-6 were significantly reduced when compared with those of PBS-treated control mice (Appendix Fig S2A). Meanwhile, IL-33 treatment markedly reduced the expression of Th1-related cytokines IFN-γ and IL-6 in allografts (Appendix Fig S2B). In contrast, IL-33 treatment enhanced the serum levels of the Th2-related cytokines IL-4 and IL-13 and the expression of IL-4 and IL-13 in allografts (Appendix Fig S2C and D). Regarding serum levels of IL-10, we found no significant differences between the IL-33-treatment group and the controls. However, IL-33 treatment markedly increased the expression of IL-10 in allografts (Appendix Fig S2). These experiments suggest that IL-33 treatment has large effects on the systemic and local Th1/Th2 response, promoting polarization of the immune response toward a Th2 phenotype, and simultaneously inhibiting Th1 reactivity. Moreover, we observed a significant increase in the ratio of CD4 T cells/CD8 T cells in islet grafts of mice treated with IL-33 (Appendix Fig S2E and F), suggesting that IL-33 treatment may prevent islet graft rejection through modulating CD4 and CD8 T-cell responses.

We and others have previously shown that IL-33 can induce expansion of regulatory T cells (Tregs) and ILC2s in multiple anatomical sites where they display an immunosuppressive role in various disease conditions (Turnquist et al, 2011; Schiering et al, 2014; Matta et al, 2016; Cao et al, 2018). Here, we aimed to examine whether short-term IL-33 treatment can induce long-term accumulation of Tregs and ILC2s in multiple anatomical sites. Firstly, we analyzed the effect of IL-33 on Tregs and ILC2s in spleen and kidney of normal C57BL/c mice at different time points after IL-33 administration. Analysis of leukocytes isolated from the spleen and kidneys of C57BL/6 mice 3 days after short-term IL-33 treatment showed a

**Figure 1. IL-33 prolonged islet allograft survival.**

A Streptozotocin-induced diabetic C57BL/6 (H2b) mice were treated with mouse recombinant IL-33 daily for 5 consecutive days before islet transplantation. On day 0, mice were transplanted with BALB/c (H2d) islets. Mice were sacrificed at day 80 post-islet transplantation or at the day when grafts were considered rejected after two consecutive BGLs > 16 mmol/l (mM) after a period of normoglycemia.

B Islet graft survival of mice receiving vehicle (PBS) or IL-33 was assessed by monitoring blood glucose and calculated using the Kaplan–Meier method. Cumulative data from two independent experiments are shown. Statistical analysis was performed with a log-rank test. ***$P$ < 0.001 vs. islet+vehicle.

C Blood glucose level of mice treated with IL-33 (the horizontal black line indicates a BGL of 16 mmol/l, the threshold for rejection). Each line represents one mouse.

D Intraperitoneal glucose tolerance test (IPGTT) was assessed in normal mice, islet transplant mice receiving vehicle (on the day when grafts were considered rejected), and islet transplant mice treated with IL-33 (at day 30 and day 80 post-islet transplantation). Data shown are the mean ± SEM ($n$ = 6–9 per group).

E Area under the curve (AUC) for IPGTT was assessed. Data shown are the mean ± SEM ($n$ = 6–9 per group), and a one-way ANOVA was performed, ***$P$ < 0.001.

F Representative immunohistochemical staining for insulin in graft samples from mice receiving vehicle or IL-33. Scale bar = 100 μm.

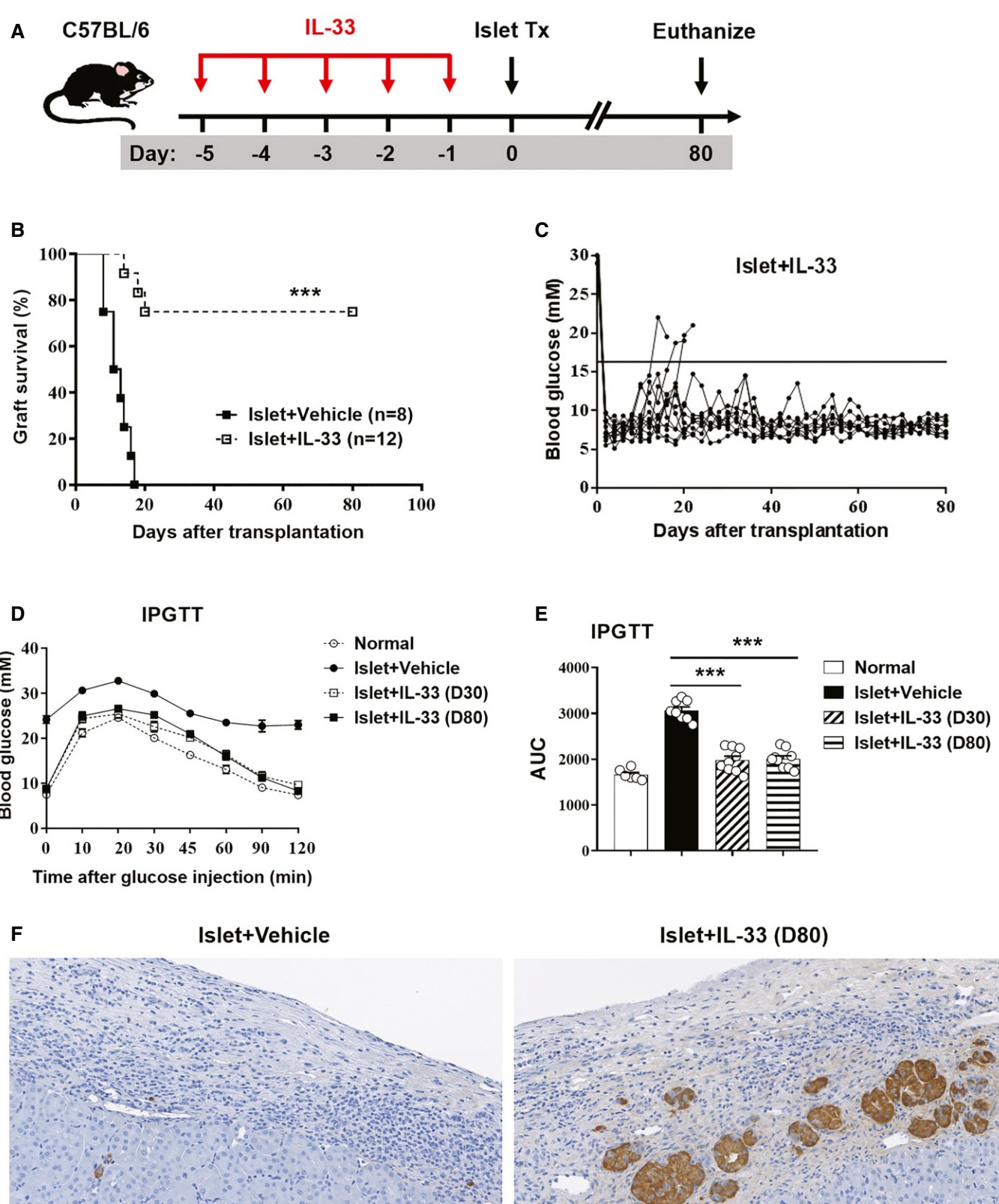

**Figure 1.**

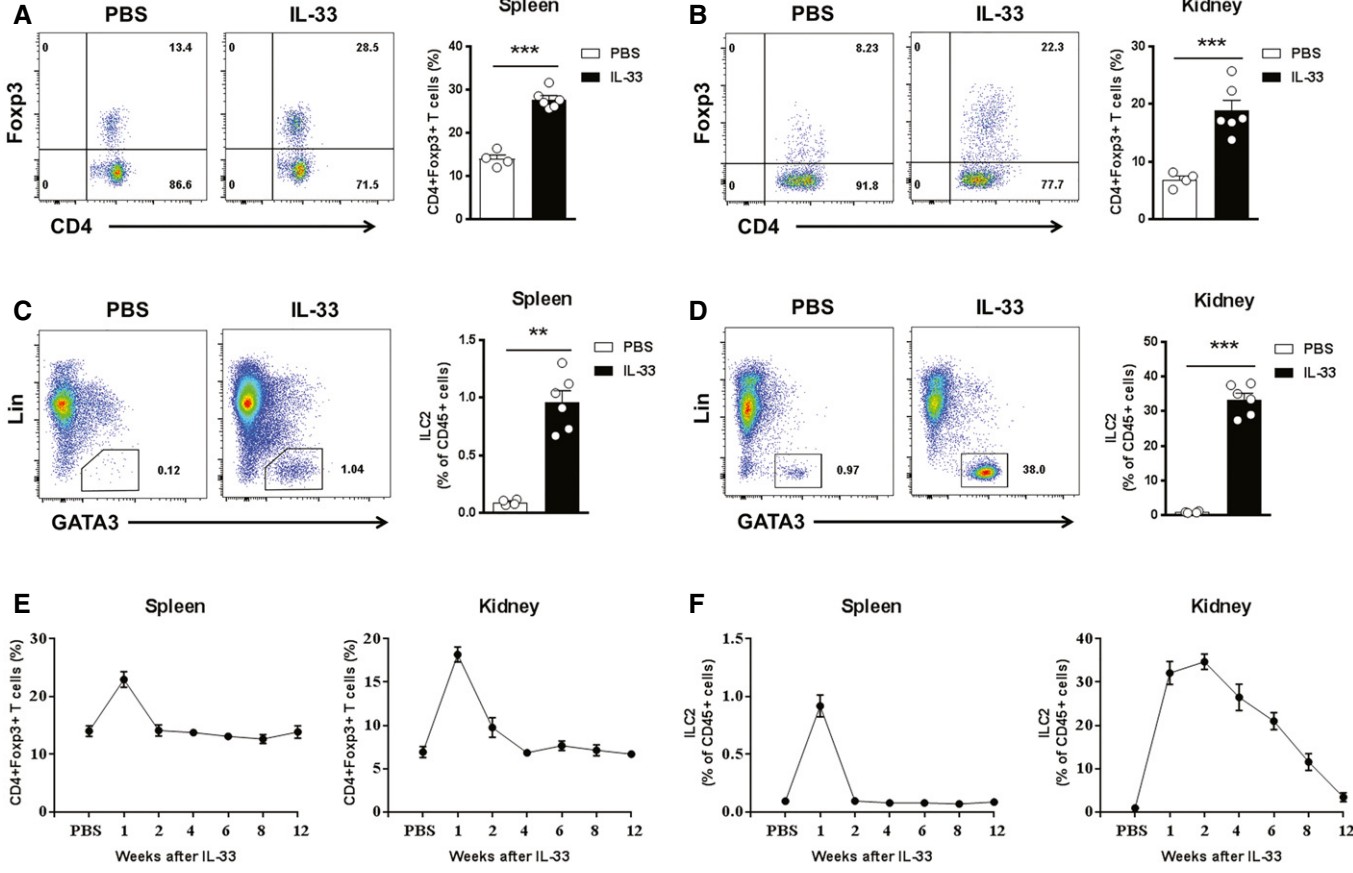

**Figure 2.   IL-33 induced Tregs and ILC2s *in vivo*.**

C57BL/6 mice were treated with mouse recombinant IL-33 or PBS daily for 5 consecutive days.

A, B   Representative FACS analysis showing the proportion of Tregs (CD4+Foxp3+) in the CD4+T-cell compartment from the spleens (A) and kidneys (B) at day 3 after treatment in C57BL/6 mice receiving PBS (*n* = 4) or IL-33 (*n* = 6).

C, D   Representative FACS analysis showing the proportion of ILC2s (Lin-GATA-3+) in the CD45+leukocyte compartment from the spleens (C) and kidneys (D) at day 3 after treatment in C57BL/6 mice receiving PBS (*n* = 4) or IL-33 (*n* = 6).

E, F   proportion of Tregs (E) and ILC2s (F) in the spleen and kidney in PBS-injected controls (*n* = 4) and at weeks 1–12 after IL-33 treatment (*n* = 4–6 per IL-33–treated group).

Data information: Data shown are the mean ± SEM; statistical analysis was performed with an unpaired *t*-test, **$P < 0.01$, ***$P < 0.001$.

moderate increase in CD4$^+$Foxp3$^+$Treg frequencies (Fig 2A and B) and a massive increase in Lin(−)GATA-3$^+$ ILC2 frequencies (Fig 2C and D) as compared with PBS-treated control mice. IL-33-induced Treg accumulation in the spleen and kidney was only maintained for 1 week after a single course of five IL-33 injections (Fig 2E),

whereas IL-33-induced ILC2 accumulation in the kidney was maintained at a high level for up to 8 weeks and the ILC2 increase was more transient in the spleen (Fig 2F). Furthermore, we examined the Tregs and ILC2 accumulation in spleen, kidney, and islet graft of islet transplant mice at different time points after IL-33

**Figure 3.   IL-33 induced Tregs and ILC2s in mice with islet transplantation.**

A   Streptozotocin-induced diabetic C57BL/6 (H2b) mice with IL-33 treatment were transplanted with BALB/c (H2d) islets. Mice were sacrificed at day 7, 30, and 80 post-islet transplantation.

B–D   Proportion or numbers of CD4+Foxp3+Tregs in the spleens, kidneys, and islet grafts of normal, islet transplant mice receiving vehicle and islet transplant mice with IL-33 treatment (at day 7, 30, and 80 post-islet transplantation). Data shown are the mean ± SEM (*n* = 4–6 per group), and a one-way ANOVA was performed; ***$P < 0.001$.

E   Representative confocal microscopy images of immunostaining for CD4, Foxp3, and insulin in islet grafts. Scale bar = 50 μm.

F–H   Proportion or numbers of ILC2s in the spleens, kidneys, and islet graft of mice with or without IL-33 treatment. Data shown are the mean ± SEM (*n* = 4–6 per group), and a one-way ANOVA was performed; **$P < 0.01$, ***$P < 0.001$.

I   Representative confocal microscopy images of immunostaining for CD127, ST2, CD3, and insulin in islet grafts. Scale bar = 50 μm.

J   The mRNA expression of IL-25, IL-33, and TSLP in islet grafts of mice with or without IL-33 treatment was examined by qPCR, and expressed relative to the control of each experiment. Data shown are the mean ± SEM (*n* = 4–6 per group), and a one-way ANOVA was performed; **$P < 0.01$, ***$P < 0.001$.

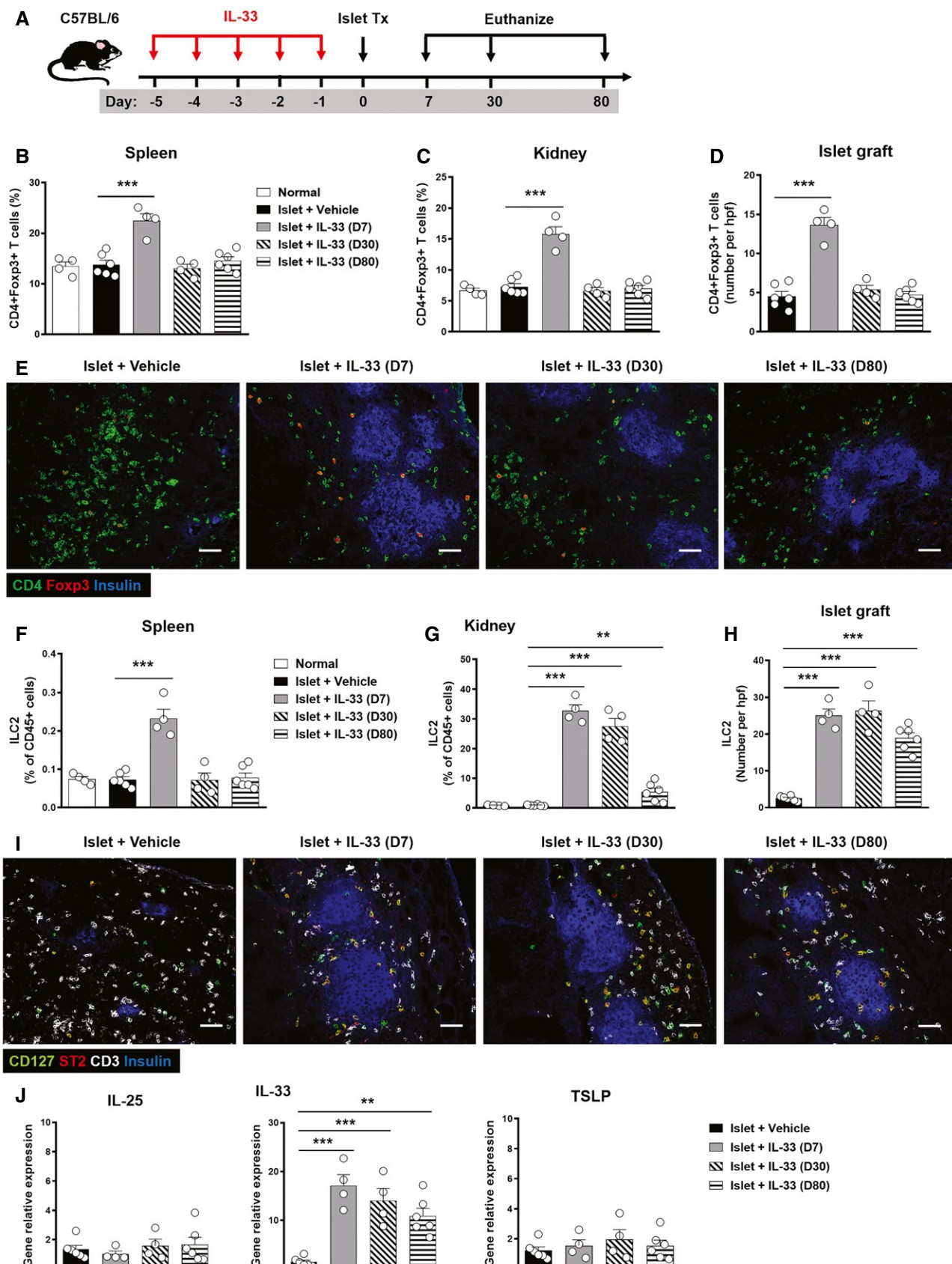

**Figure 3.**

administration (Fig 3A). As expected, IL-33 treatment induced the accumulation of Tregs in spleen, kidney, and islet graft at day 7 post-islet transplantation, but not at day 30 and day 80 (Fig 3B–D), indicating IL-33 induced only a short-term increase in Tregs in different tissues, especially islet graft. In contrast, IL-33 treatment induced a long-term increase in ILC2s in kidney and islet graft, but not in the spleen (Fig 3E–I). A greater amount of ILC2s were found in islet graft than in kidney and liver, which indicates that ILC2s tend to migrate to islet graft undergoing immune response (Appendix Fig S3). Moreover, we observed a consistent increase in IL-33, but not IL-25 or thymic stromal lymphopoietin (TSLP), in islet graft tissue of mice treated with IL-33 (Fig 3J). These data could partly explain why ILC2s are found within the graft for so long. Taken together, a short course of IL-33 treatment induced a sustained increase in ILC2 abundance in kidney and islet graft which may be involved in IL-33-mediated islet graft protection.

## Tregs and ILC2s played critical roles in IL-33-mediated islet graft protection

Previous studies have shown that *ex vivo*-expanded Tregs protected against islet graft rejection (Shi *et al*, 2012; Yi *et al*, 2012). IL-33-mediated cardiac allograft survival and acute graft-versus-host disease (GVHD) protection was dependent on Tregs (Turnquist *et al*, 2011; Matta *et al*, 2016). To examine whether Treg accumulation *in vivo* contributes to IL-33-mediated islet graft protection, depletion of regulatory T cells (DEREG) mice was administered diphtheria toxin (DT) to selectively deplete Tregs *in vivo* (Fig 4A). Treg depletion in DEREG mice was confirmed in the spleen and kidney by flow cytometry at day 5 post-islet transplantation (Fig 4B). Treg depletion significantly reduced the survival rate of islet graft in IL-33-treated mice, suggesting that Treg accumulation is an important mechanism in IL-33-mediated islet graft protection (Fig 4D and E). ILC2s play important roles in tissue repair and immunoregulation (Sonnenberg & Artis, 2015). We and other groups found that IL-33-activated ILC2s expressed significantly higher levels of CD25 *in vivo* (Appendix Fig S4; Roediger *et al*, 2015), suggesting that administration of anti-CD25 antibodies could be utilized to deplete ILC2s *in vivo*. However, administration of anti-CD25 antibody has been used to successfully deplete Tregs *in vivo* as CD25 is also highly expressed on Tregs (Lu *et al*, 2013). Here, the potential contribution of ILC2s to IL-33-mediated islet graft protection was assessed in Treg-depleted mice (DEREG mice treated with DT), in which administration of anti-CD25 antibodies (PC61) leads to depletion of ILC2s (Fig 4A). Flow cytometric analysis of Lin-GATA3[+] ILC2s in spleen and kidney confirmed significant depletion

of ILC2s in mice with anti-CD25 antibody treatment compared with Treg-depleted DEREG mice (Fig 4C). Additional ILC2 depletion in Treg-depleted DEREG mice completely abolished the protective effects of IL-33 on islet transplantation (Fig 4D and E). Taken together, these results demonstrate that both Tregs and ILC2s play critical roles in IL-33-mediated islet graft protection.

## IL-33 and IL-2/anti-IL-2 antibody complex induced ILC2[10]

IL-10 producing ILC2s (ILC2[10]) have been described in lung and intestine (Seehus *et al*, 2017; Wang *et al*, 2017). IL-33 and IL-2 have been shown to induce ILC2[10] *in vivo* (Seehus *et al*, 2017). Here, we identified two subsets of ILC2s in islet graft and kidney of islet transplant mice treated with IL-33, ILC2[10], and non-IL-10 producing ILC2s (Fig 5A and B). The IL-2/anti-IL-2 antibody complex (IL-2C) has been found to directly induce expansion of ILC2 that express the high-affinity IL-2 receptor CD25 (Seehus *et al*, 2017; Cao *et al*, 2020). Using IL-10 reporter mice, we further demonstrated that combined IL-33 and IL-2C treatment markedly enhanced the ILC2[10] expansion in the kidney when compared with IL-33 treatment alone (Fig 5C). *In vitro*, ILC2s isolated from kidneys were treated with IL-33 and IL-2C and analyzed for IL-10 by flow cytometry and ELISA. IL-33 and IL-2C treatment induced more ILC2[10] expansion in cultured ILC2s (Fig 5D). ILC2s when cultured with IL-33 and IL-2C produced a high level of IL-10 in the supernatant (Fig 5E). IL-2 is known to activate STAT5 which has been shown to promote IL-10 expression in T cells (Polhill *et al*, 2012). Therefore, we evaluated the degree of phosphorylation of STAT5 (p-STAT5) in ILC2s. As expected, kidney ILC2s treated with IL-33 and IL-2C rapidly increased the expression of p-STAT5 *in vitro* (Fig 5F). Moreover, inhibition of STAT5 significantly reduced IL-10 production by ILC2s cultured with IL-33 and IL-2C (Fig 5G).

## ILC2[10] prolonged islet graft survival in an IL-10-dependent manner

ILC2[10] produced large amounts of IL-10, an anti-inflammatory cytokine, which has been demonstrated to suppress graft rejection in different transplant models (Gagliani *et al*, 2010; Yi *et al*, 2012). To assess the importance of IL-10 for ILC2[10] cell function *in vivo*, we deleted IL-10 in ILC2[10] using CRISPR-Cas9. ILC2[10] transfected with control empty vector produced a large amount of IL-10 in the supernatant, whereas ILC2[10] transfected with IL-10 CRISPR-Cas9 did not (Fig 6A). We then adoptively transferred ILC2[10], IL-10-deleted ILC2[10], and non-ILC2[10] into islet transplant C57BL/6 mice (Fig 6B). Fluorescently labeled ILC2[10], IL-10-deleted ILC2[10], and non-ILC2[10]

---

**Figure 4. ILC2s and Tregs contributed to IL-33-mediated islet protection *in vivo*.**

A   Streptozotocin-induced diabetic DEREG C57BL/6 mice were treated with mouse recombinant IL-33 daily for 5 consecutive days, as well as diphtheria toxin (DT), PC61 or DT+PC61 on days −4 and sletallogr1 prior to and on day 2 post-islet transplantation. Mice were sacrificed at day 80 post-islet transplantation or at the day when grafts were considered rejected.

B, C  Proportion of CD4[+]Foxp[+]Tregs (B) and Lin-GATA3[+]ILC2s (C) from the spleens and kidneys of islet transplant mice receiving vehicle, IL-33, IL-33/DT, IL-33/PC61, or IL-33/DT/PC61 at day 5 post-islet transplantation. Data shown are the mean ± SEM ($n$ = 4–5 per group), and a one-way ANOVA was performed; NS: non-significant, ***$P$ < 0.001.

D   Islet graft survival of five groups of mice was assessed by monitoring blood glucose and calculated using the Kaplan–Meier method. Cumulative data from two independent experiments are shown. Statistical analysis was performed with a log-rank test. *$P$ < 0.05, **$P$ < 0.01.

E   Data are shown as blood glucose measurement in islet transplant mice treated with IL-33/DT, IL-33/PC61, or IL-33/DT/PC61. The horizontal black line indicates a BGL of 16 mmol/l, the threshold for rejection. Each line represents one mouse.

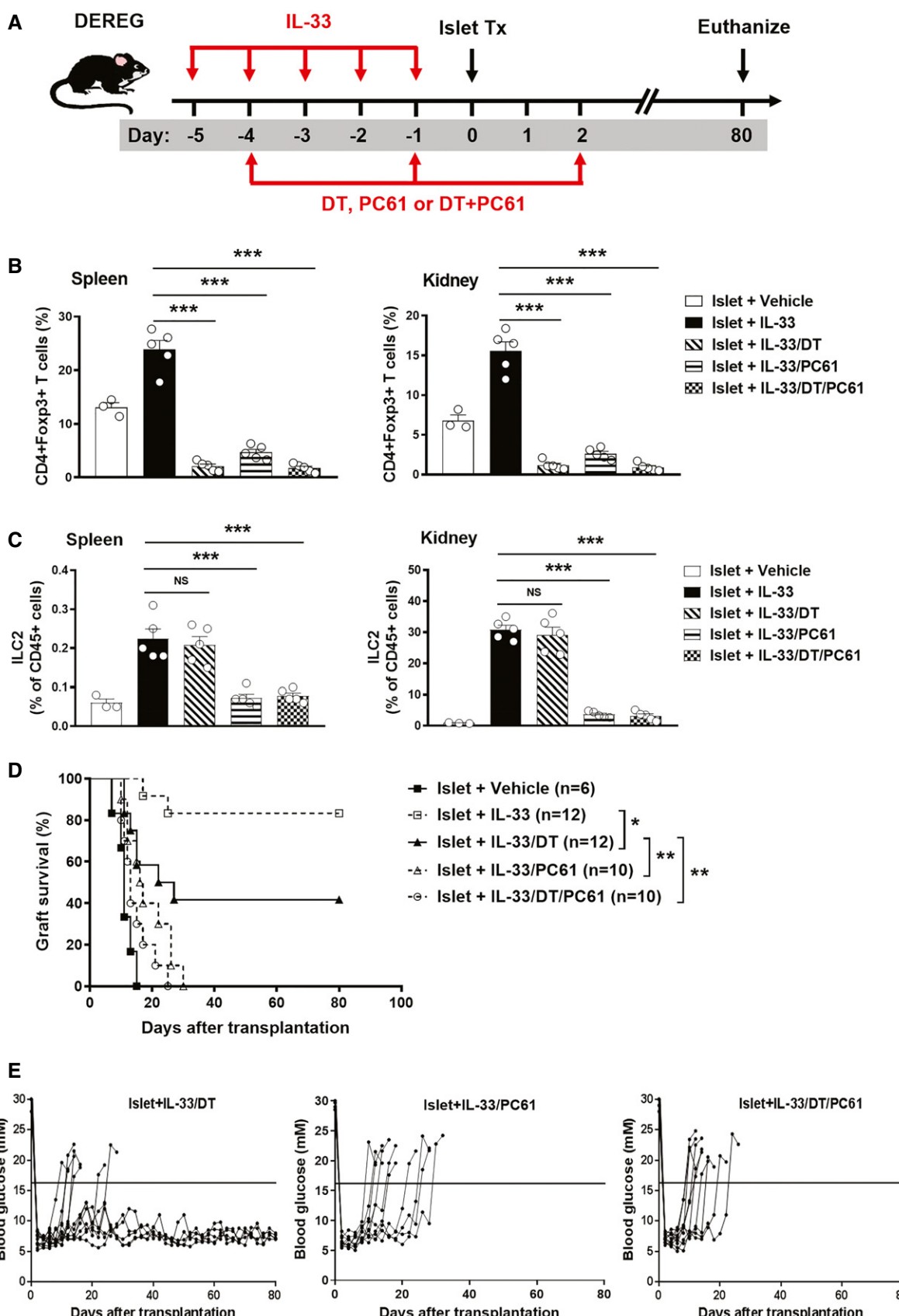

**Figure 4.**

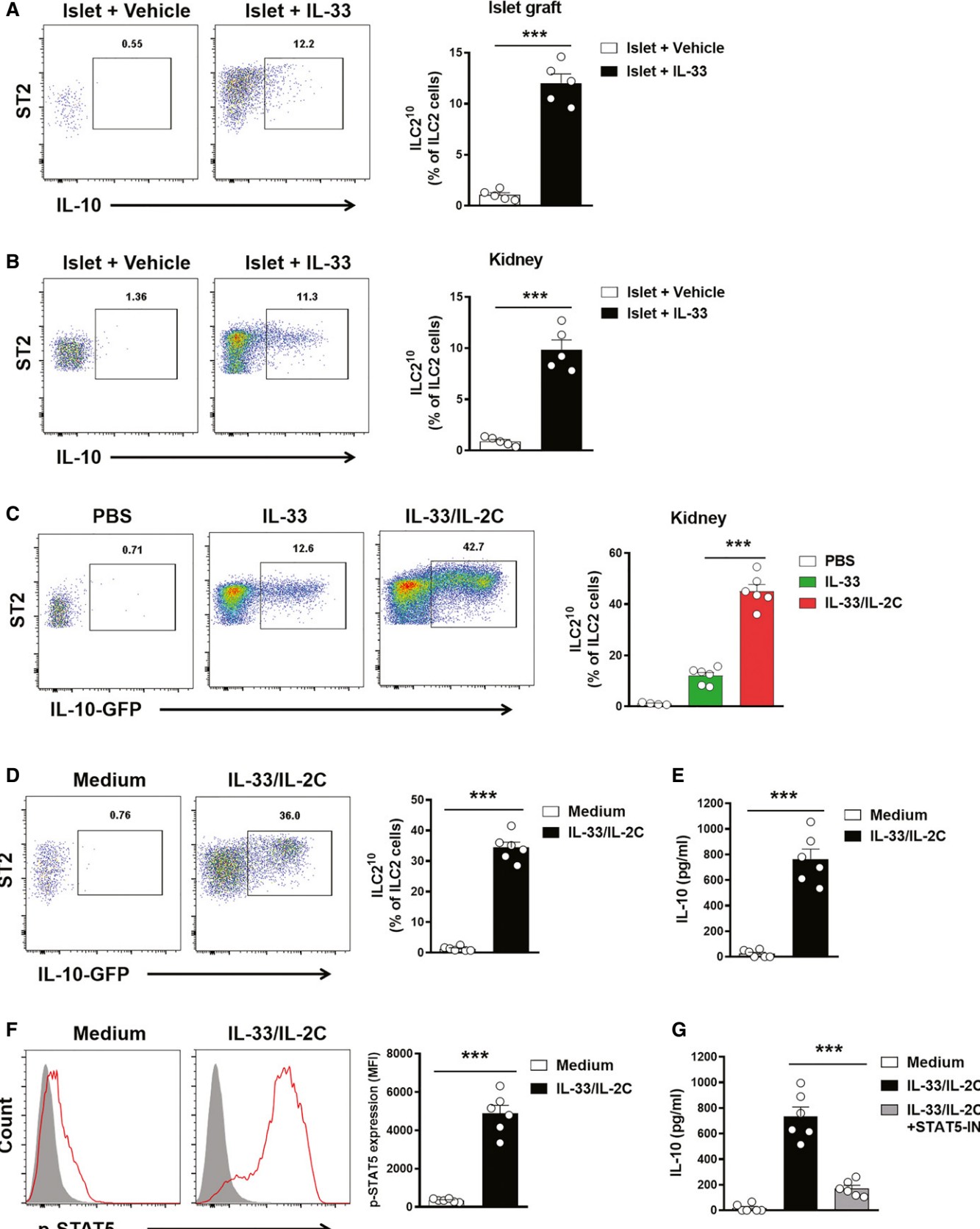

Figure 5.

**Figure 5. IL-33 and IL-2C induced ILC2[10].**

A, B  ILC2[10] and non-IL-10 producing ILC2s were assessed in islet graft (A) and kidney (B) by intracellular IL-10 staining at day 5 post-islet transplantation. Data shown are the mean ± SEM (n = 5 per group), and an unpaired *t*-test was performed; ***P < 0.001. ST2: suppression of tumorigenicity 2.

C  IL-10 reporter C57BL/6 mice were treated with PBS, IL-33 alone or IL-33 and IL-2C daily for 5 consecutive days. Proportion of lin-CD127+ST2+IL-10-GFP ILC2[10] from the kidneys at day 3 after treatment in IL-10 reporter C57BL/6 mice. Data shown are the mean ± SEM (n = 4–6 per group), and a one-way ANOVA was performed; ***P < 0.001.

D  Kidney ILC2s isolated from normal IL-10 reporter C57BL/6 mice were cultured with medium only or medium with IL-33 and IL-2C for 6 days. Proportion of ILC2[10] was assessed by flow cytometry. Data shown are the mean ± SEM (n = 6 per group), and an unpaired *t*-test was performed; ***P < 0.001. ST2: suppression of tumorigenicity 2.

E  The secreted cytokine IL-10 was analyzed via ELISA. Data shown are the mean ± SEM (n = 6 per group), and an unpaired *t*-test was performed; ***P < 0.001.

F  Kidney ILC2s isolated from C57BL/6 mice were cultured with medium only or medium with IL-33 and IL-2C for 30 min. The expression of phosphorylated STAT5 was examined in kidney ILC2s by flow cytometry. p-STAT5 (red lines) and isotype controls (gray-filled areas) are shown. Data represent the mean ± SEM of evaluations of MFI from each group (n = 6 per group), and an unpaired *t*-test was performed; ***P < 0.001. MFI, mean fluorescence intensity.

G  The kidney ILC2s were cultured with medium, IL-33/IL-2C, or IL-33/IL-2C and STAT5 inhibitor (STAT5-IN) for 3 days. The secreted cytokine IL-10 was analyzed via ELISA. Data shown are the mean ± SEM (n = 6 per group), and a one-way ANOVA was performed; ***P < 0.001.

were observed in islet grafts of islet transplant mice, and there was no difference in the number of transfused ILC2s among the three groups (Fig 6C). The expression of IL-10 in islet graft was significantly increased in mice transfused with ILC2[10], but not in mice transfused with IL-10-deleted ILC2[10] or non-ILC2[10] (Fig 6D). There was no difference in IL-10 levels in plasma among the four groups (data not shown). ILC2[10] treatment significantly prolonged islet allograft survival, whereas non-ILC2[10] and IL-10-deleted ILC2[10] did not provide protection (Fig 6E). These results indicated that ILC2[10] prolong islet allograft survival via production of IL-10. The immunosuppressive effect of ILC2[10] on CD4 T cells was assessed by co-culture *in vitro*. ILC2[10] effectively suppressed allogeneic splenocyte-induced CD4 T-cell proliferation in a dose-dependent manner, and neutralizing anti-IL-10 antibody or genetic ablation of IL-10 diminished the suppressive role of ILC2[10] on CD4 T-cell proliferation (Fig 6F and G). Therefore, these results demonstrated that IL-10 is an important mediator in ILC2[10]-mediated islet allograft survival.

## Co-transplantation of ILC2[10] with islet prolonged allograft survival

We next determined whether migration to islet allografts was important for ILC2[10] cell effector function. To explore this, we compared the protective effects of intravenous and local transfer of ILC2[10] cells on islet rejection (Fig 7A). Intravenous transfer of 2 × 10^6 ILC2[10] cells modestly prolonged graft survival from 10.4 ± 2.5 days to 36.3 ± 4.5 days, whereas locally transferred 1 × 10^6 ILC2[10] cells led to long-term islet graft survival—37.5% (n = 3 out of 8) of the mice retained their grafts for more than 80 days. Local transfer of 1 × 10^6 ILC2[10] cells and 2 × 10^5 ILC2[10] cells exhibited similar islet allograft protection, whereas local transfer IL-10-deleted ILC2[10] did not prolong islet graft survival (Fig 7B). Furthermore, we locally transferred CD45.2+ ILC2[10] cells into islet transplanted congenic C57BL/6 mice (CD45.1+) and examined their frequency in islet graft over time. The locally transferred CD45.2+ST2+ILC2[10] were detected in islet graft at day 5 post-islet transplantation, and their proportion and numbers in islet graft were significantly reduced at the day when grafts were rejected or at day 80 post-islet transplantation (Fig 7C–E). The proportion of CD45.2+ST2+ILC2[10] in accepted islet graft (surviving for 80 days post-islet transplantation) was higher than that in rejected islet graft, but there was no difference in the number of CD45.2+ST2+ILC2[10] in rejected islet graft and in accepted islet graft (surviving for 80 days post-islet transplantation) (Fig 7D and E),

suggesting that the occurrence of islet rejection is not because the number of ILC2 reduces in the graft over time. Phenotypic alteration of ILC2 has been reported in various inflammatory disease conditions (Ohne *et al*, 2016; Golebski *et al*, 2019); here, we found that the expression of IL-10, IL-5, and IL-13 by locally transferred ILC2[10] (Fig 7F and Appendix Fig S5) was markedly reduced in mice with islet graft rejection in comparison with that in mice that retained their islet graft for 80 days. Phenotypic change in ILC2[10] could possibly explain why locally transferred ILC2[10] did not lead to long-term islet graft survival in 5 out of 9 islet transplanted mice. There results suggested that ILC2[10] cells were required within the allograft for maximal suppressive effect and graft protection.

## Discussion

A growing body of evidence suggests ILC2s play immune regulatory roles in acute and chronic inflammatory diseases (Rak *et al*, 2016; Dalmas *et al*, 2017; Gadani *et al*, 2017; Cao *et al*, 2018; Krabbendam *et al*, 2018), whereas their possible importance in suppressing allograft rejection is unclear. Here, we found that short-term IL-33 treatment significantly prolonged islet allograft survival in STZ-induced type 1 diabetic mice. The mechanisms underlying the protective effect of IL-33 are associated with its initiation of suppressing alloimmune responses through expansion of ILC2s and Tregs. Furthermore, two functional subpopulations of ILC2s were identified in islet grafts of IL-33-treated mice. IL-10 producing ILC2s (ILC2[10]) were demonstrated to play a critical role in IL-33-mediated islet allograft protection.

IL-33 is a novel member of the IL-1 cytokine family and acts as a dual-function molecule, namely as a nuclear gene regulator and extracellular cytokine (Molofsky *et al*, 2015; Martin & Martin, 2016). IL-33 plays an anti-inflammatory role in atherosclerosis, cutaneous wound healing, intestinal, and adipose tissue inflammation via expansion of Tregs (Miller *et al*, 2008; Schiering *et al*, 2014; Brestoff *et al*, 2015; Monticelli *et al*, 2015; Rak *et al*, 2016). Recent reports have highlighted that Tregs contribute to the protective effect of IL-33 in models of cardiac allograft rejection and graft-versus-host disease (Turnquist *et al*, 2011; Matta *et al*, 2016). In this study, we demonstrated for the first time that IL-33 promoted long-term islet allograft survival in a fully MHC mismatched murine islet transplantation model. Tregs were significantly increased in spleen, kidney, and allograft of IL-33-treated mice. Tregs exhibit potent anti-inflammatory

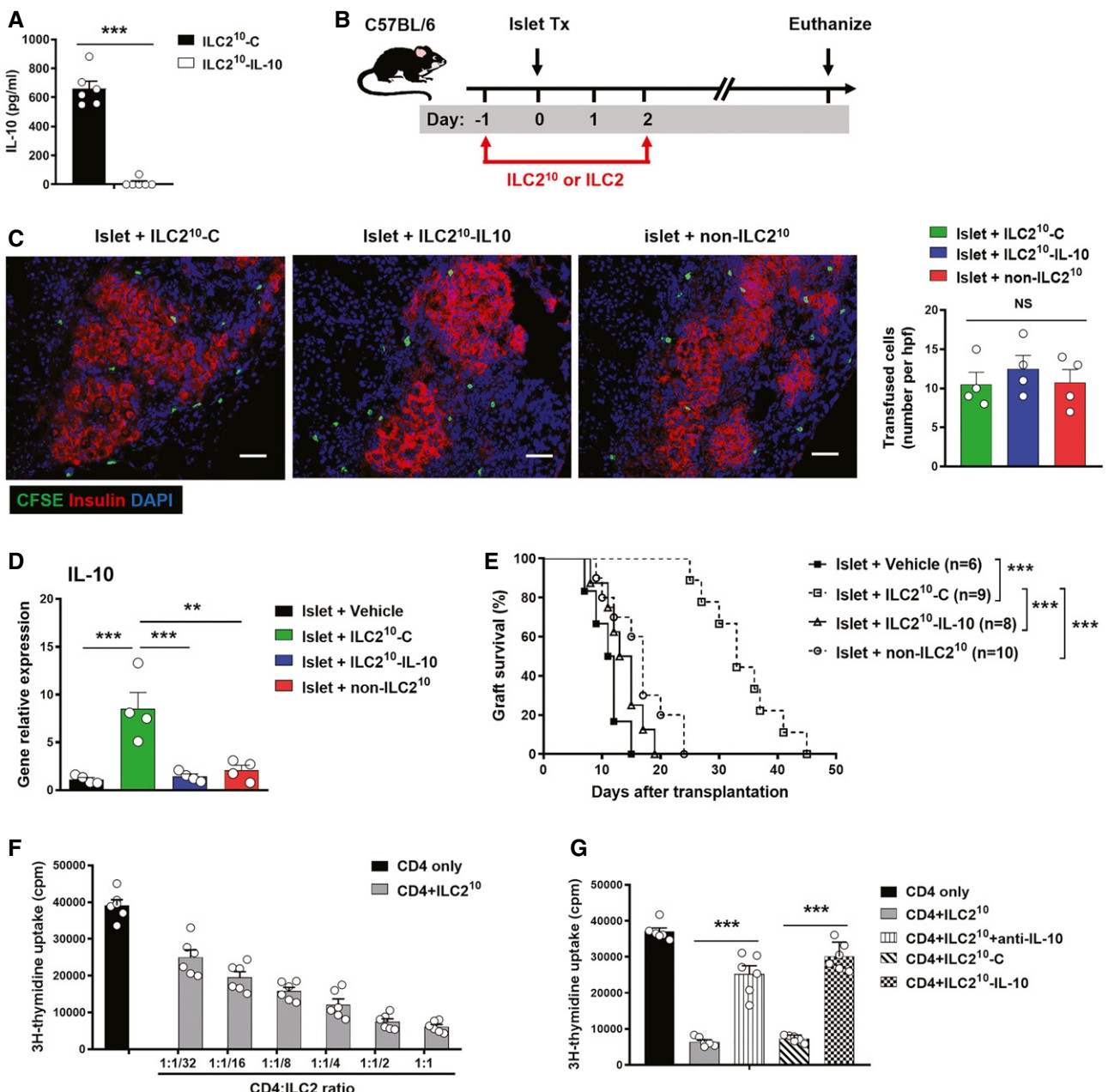

**Figure 6. Transfused ILC2^10 suppressed islet rejection via IL-10.**

A ILC2^10 and non-ILC2^10 were isolated from *ex vivo*-expanded kidney ILC2 by flow sorting. ILC2^10 were transfected with control (ILC2^10-C) or IL-10 CRISPR-Cas9 (ILC2^10-IL-10). IL-10 was measured in culture supernatant of ILC2^10-C and ILC2^10-IL-10 via ELISA. Data shown are the mean ± SEM (*n* = 6 per group), and an unpaired *t*-test was performed; ***P* < 0.001.

B Transfected ILC2^10-C and ILC2^10-IL-10, and non-ILC2^10 were adoptively transferred into diabetic C57BL/6 mice twice at 1 day prior to and 2 days post-islet transplantation. Mice were sacrificed at the day when grafts were considered rejected.

C Transfused ILC2s (CFSE labeled, Green) were observed in islet graft at day 5 post-islet transplantation. Scale bar = 50 μm. The numbers of CFSE-labeled ILC2s in the islet graft were counted. Data shown are the mean ± SEM, and a one-way ANOVA was performed (*n* = 4 per group). NS, non-significant.

D The mRNA expression of IL-10 in islet grafts at day 5 post-islet transplantation was examined by qPCR. Data shown are the mean ± SEM (*n* = 4 per group), and a one-way ANOVA was performed; ***P* < 0.01, ****P* < 0.001.

E Islet graft survival of four groups of mice was assessed by monitoring blood glucose and calculated using the Kaplan–Meier method. Cumulative data from two independent experiments are shown. Statistical analysis was performed with a log-rank test. ****P* < 0.001.

F CD4^+ T cells isolated from C57BL/6 splenocytes were cultured with irradiated splenocytes derived from BALB/c in the presence of ILC2^10 at the indicated ratios for 4 days. CD4^+ T-cell proliferation was assessed using [^3H]-thymidine incorporation assays. Data shown are the mean ± SEM (*n* = 4–6 per group).

G Neutralizing anti-IL-10 antibodies or genetic ablation of IL-10 were used to block the effect of ILC2^10 on CD4^+ T-cell proliferation. Data shown are the mean ± SEM (*n* = 6 per group), and an unpaired *t*-test was performed. ****P* < 0.001.

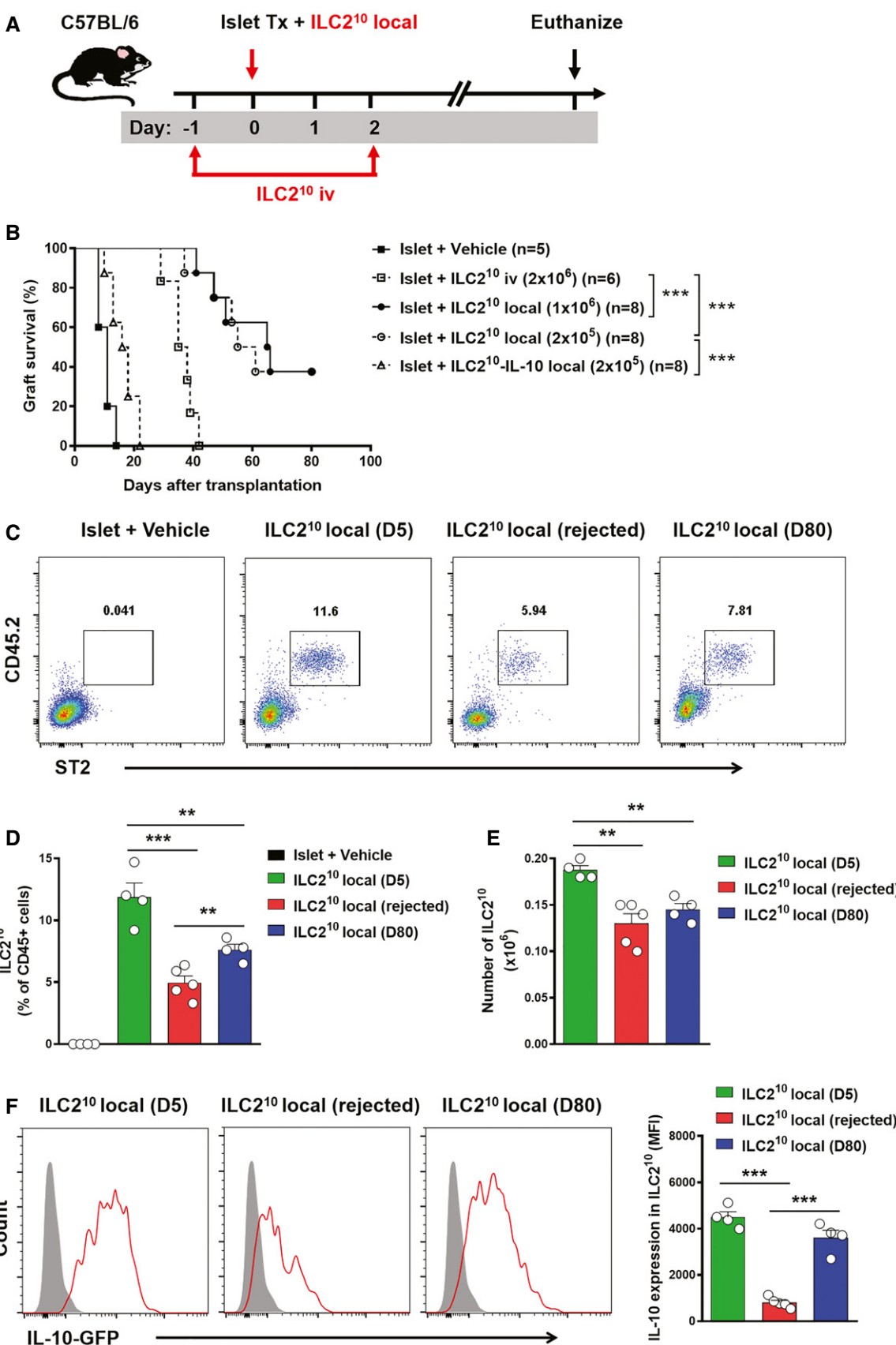

**Figure 7.**

◄

**Figure 7. Co-transplantation of IL-2[10] with islet prolonged allograft survival.**

A    ILC2[10] were isolated from *ex vivo*-expanded kidney ILC2s by flow sorting. ILC2[10] were co-transplanted with islet locally or adoptively transferred into diabetic C57BL/6 mice twice at 1 day prior to and 2 days post-islet transplantation. Mice were sacrificed at day 80 post-islet transplantation or at the day when grafts were considered rejected.

B    Islet graft survival of five groups of mice was assessed by monitoring blood glucose and calculated using the Kaplan–Meier method. Cumulative data from two independent experiments are shown. Statistical analysis was performed with a log-rank test. ***$P < 0.001$.

C    To track ILC2[10] *in vivo*, CD45.2[+]ILC2[10] were co-transplanted with islets in diabetic CD45.1[+]C57BL/6 mice. Representative FACS analysis showing the proportion of locally transplanted ILC2[10] (CD45.2[+]ST2[+]) in the total CD45[+] cell compartment from islet graft at day 5 and 80 post-islet transplantation or at the day when grafts were considered rejected.

D, E  Proportion and numbers of locally transplanted CD45.2[+]ST2[+]ILC2[10] in islet grafts of islet transplant mice over time. Data shown are the mean ± SEM ($n$ = 4–5 per group), and a one-way ANOVA was performed, **$P < 0.01$, ***$P < 0.001$.

F    The expression of IL-10-GFP was examined in locally transplanted CD45.2[+]ST2[+]ILC2[10] by flow cytometry. IL-10-GFP (red lines) and controls (gray-filled areas) are shown. Data represent the mean ± SEM of evaluations of MFI from each group ($n$ = 4–5 per group), and a one-way ANOVA was performed; ***$P < 0.001$. MFI, mean fluorescence intensity.

properties and play an immunosuppressive role in transplant rejection (Issa *et al*, 2010; Yi *et al*, 2012). As expected, an increase in Tregs contributes to IL-33-mediated allograft survival, which is consistent with previous studies showing that *ex vivo*-expanded Tregs can prolong islet allograft survival (Shi *et al*, 2012; Yi *et al*, 2012). Furthermore, ILC2s were shown to be crucial for the function of IL-33 in promoting islet allograft survival, as additional ILC2 depletion in Treg-depleted DEREG mice counteracted any therapeutic effect of the IL-33 treatment. We also examined the spatial and temporal distribution of Tregs and ILC2s induced by IL-33 treatment. A short course of IL-33 treatment induced a sustained increase in ILC2s in kidney and islet graft, but only a short-term increase in Treg in those tissues, suggesting ILC2s accumulated in islet allografts to play critical roles in IL-33-mediated islet graft protection. Interesting, the reason why ILC2s stay longer than Tregs may be related to preservation of a high concentration of IL-33 in islet graft that extends the life of ILC2s. Another potential mechanism of IL-33-mediated islet graft protection is associated with induction of a Th1-to-Th2 switch in the periphery and allograft, which has been shown to contribute to IL-33-mediated cardiac allograft survival (Turnquist *et al*, 2011).

In further exploring diversity of ILC2 lineage, we found that ILC2s can be divided into two subsets: one producing IL-10 (ILC2[10]) and another not producing IL-10 (non-ILC2[10]). An important finding in the current study is that ILC2[10], but not non-ILC2[10], play a critical role in islet allograft survival. We have previously reported that IL-33 induced significant expansion of ILC2s in kidney, which play a protective role in renal IRI (Huang *et al*, 2015; Cao *et al*, 2018). As ILC2s expressed high level of receptors for IL-2 and IL-33, we examined whether the combination of IL-2/anti-IL-2 antibody complex (IL-2C) and IL-33 treatment would influence generation of ILC2[10] *in vivo*. We found that the administration of IL-2C and IL-33 significantly increased the ILC2[10] population in kidney, which is consistent with an earlier report in lung (Seehus *et al*, 2017). IL-10 is a well-characterized anti-inflammatory cytokine and plays a central role in controlling inflammation and suppressing alloimmune responses after transplantation (Hara *et al*, 2001; Ouyang & O'Garra, 2019). The importance of IL-10 in Treg or Tr1-mediated suppression of alloimmune responses has been demonstrated in many transplantation models (Gagliani *et al*, 2010; Yi *et al*, 2012). ILC2[10] produced large amounts of the anti-inflammatory cytokine IL-10, which could suppress the alloimmune response to allogeneic islets. This is supported by our data showing that ILC2[10] effectively suppressed allogeneic splenocyte-induced CD4 T-cell proliferation via secretion of IL-10 and transfused ILC2[10] prolonged islet graft

survival in an IL-10-dependent manner. Moreover, another important finding is that local transfer of ILC2[10] together with islet exhibited a stronger protective effect on allogeneic islets than systemic transfer of ILC2[10] cells. Also, the phenotypic stability of locally transferred ILC2[10] is an important factor in determining the fate of islet grafts. Similar to our findings, Treg migration from blood to the islet allograft was necessary for the suppression of alloimmunity (Zhang *et al*, 2009). These results indicate that ILC2[10] cells are required within the allograft in order to exert maximal suppressive effect and graft protection. This also suggests that ILC2[10] can efficiently suppress alloimmune response at the site of the inflamed allograft. Furthermore, mounting evidence has shown that ILC2s directly promote would healing in models of respiratory infection, acute kidney injury, and intestinal inflammation via producing Areg (Monticelli *et al*, 2011, 2015; Cao *et al*, 2018), which allows a better re-epithelialization of the tissue. These observations are very important in the field of transplantation, where correct repair of damaged tissue is essential for proper function of the allograft. This could be another important mechanism associated with ILC2-mediated allograft protection, which needs to be further investigated.

In conclusion, this study has revealed a strong protective effect of IL-33 on the survival of islet allografts. The allograft protective effect of IL-33 depends on its ability to drive expansion of ILC2s and Tregs *in vivo*. Furthermore, this study demonstrates the importance of IL-10 in ILC2-mediated islet graft protection. Migration of ILC2[10] cells into the islet graft was necessary for maximal suppressive effect and graft protection. Local delivery of ILC2[10] could be a promising tool to promote long-term islet graft survival. Therefore, this study suggests the strategy of using IL-33 and ILC2[10] as adjunctive therapy to prevent allograft rejection, bringing new therapeutics to the transplantation field.

# Materials and Methods

### Mice

C57BL/6 (CD45.2[+]), congenic C57BL/6 (CD45.1[+]), and BALB/c mice were purchased from the Animal Resources Centre (Perth, Australia) and Shanghai Laboratory Animal Center of Chinese Academy of Science. DEREG mice (C57BL/6-Tg23.2Spar/Mmjax) (Lahl *et al*, 2007) and IL-10-GFP mice (B6.129S6-Il10tm1Flvj) (Kamanaka *et al*, 2006) were obtained from Jackson Lab and bred at the Department of Animal Care at Xinxiang Medical University

(XMU) and Westmead Hospital Animal House. For all studies, adult (8–10 weeks of age) male mice were used in accordance with the animal care and use protocols approved by Animal Ethics Committee of XMU or Western Sydney Local Health District (WSLHD).

### Islets transplantation and IL-33 treatment

C57BL/6 recipient mice were rendered diabetic using a single i.v. injection of streptozotocin (185 mg/kg). Mice with a blood glucose value > 16 mmol/l were selected as transplant recipients. Pancreatic islets were separated from the pancreata of donor (BALB/c) mice at a ratio of four pancreata per recipient. Briefly, islets were isolated from adult BALB/c mice by pancreas perfusion using Liberase TL Enzyme (Roche), digestion, and purification by Ficoll (Sigma) discontinuous gradient centrifugation. The interface between 1.081 and 1.037 was the separated islet cells. Islet transplantation was performed as previously described (Szot *et al*, 2007), kidney was exposed via a small incision in the peritoneum, a small nick was made in the kidney capsule at the inferior renal pole, and the islets (2,000 IEQ/recipient mouse) were deposited through the nick toward the superior pole of the kidney. Graft rejection was defined as a rise in blood glucose above > 16 mmol/l for two consecutive days after a period of euglycemia. Nephrectomy was performed at day 80 post-islet transplantation to determine whether the euglycemia was graft dependent.

For IL-33 treatment, C57BL/6 mice were given 0.3 μg mouse recombinant IL-33 (BioLegend) intraperitoneally daily for five consecutive days before islet transplantation. The dose and duration were selected according to previous published studies (Monticelli *et al*, 2015; Cao *et al*, 2018). Control animals received phosphate-buffered saline (PBS) only. To deplete Tregs and/or ILC2s, DEREG mice were injected i.p. with diphtheria toxin (DT; 30 mg/kg, Sigma-Aldrich) and/or anti-CD25 antibody (PC61, 500 μg/mouse, BioLegend). Tregs and ILC2s were examined in spleen and kidney by flow cytometry. Mice were sacrificed at day 80 post-islet transplantation or at the day when grafts were considered rejected after two consecutive BGLs > 16 mmol/l (mM). For STZ-induced diabetic C57BL/6 mice without islet transplantation, hyperglycemic mice were injected with IL-33 daily for 5 days (starting from day 6 after STZ injection). Fasting and non-fasting blood glucose concentrations and survival rate of diabetic mice were monitored for 30 days.

### Intraperitoneal glucose tolerance test

Intraperitoneal glucose tolerance test was performed in normal mice, islet transplant mice, and islet transplant mice treated with IL-33. After 16 h overnight fasting, mice were injected with 1 g/kg body weight of glucose intraperitoneally. Blood glucose was measured at the indicated minutes after glucose injection.

### In vitro suppression assay

CD4$^+$ T cells isolated from C57BL/6 splenocytes (responders; $1 \times 10^5$ cells/well) were cultured in round-bottom 96-well plates with irradiated splenocytes (stimulators; $2 \times 10^5$/well) derived from BALB/c in the presence of ILC2$^{10}$ at various ratios for 4 days. Neutralizing anti-IL-10 antibodies (10 μg/ml, JES5-16E3, BioLegend)

were used to block the effect of ILC2$^{10}$ on CD4$^+$ T cells proliferation. Cells were pulsed with 1 μCi [$^3$H]thymidine per well for the last 16 h of a 4-day culture period. Cells were harvested using a Packard Filtermate Harvester 96 and counted by Microbeta counter (PerkinElmer).

### CRISPR-Cas9 transfection

To knock out IL-10 gene in ILC210, we co-transfected IL-10 CRISPR-Cas9 KO plasmid with an IL-10 HDR plasmid (Santa Cruz Biotechnology), following the manufacturer's instructions. In brief, ILC210 were transfected with IL-10 CRISPR-Cas9 plasmid or its control (sc-421076) and incubated for 24 h. Media were replaced 24 h post-transfection. Puromycin antibiotic (2 μg/ml) was added to allow for positive selection of transfected cells. IL-10 was measured in culture supernatant of ILC2$^{10}$ via enzyme-linked immunosorbent assay (ELISA). ELISA was performed according to the manufacturer's protocol (eBioscience). Briefly, pre-coated microtiter plates were blocked at room temperature for 2 h with PBS containing 2% BSA. Each sample and its control were added to adjacent wells and incubated overnight at 4°C. After washing, avidin-conjugated HRP and tetramethylbenzidine were used for color development. Optical densities were measured using an ELISA reader, and the concentration of cytokine was calculated.

### ILC2$^{10}$ cell isolation and administration in islet transplant mice

ILC2$^{10}$ and non-ILC2$^{10}$ were isolated from kidney ILC2s in IL-10-GFP C57BL/6 mice treated with IL-33 and IL-2C daily for five consecutive days. To examine *in vivo* functions of ILC2$^{10}$ in islet transplant mice, ILC2$^{10}$-C ($1 \times 10^6$), ILC2$^{10}$-IL-10 ($1 \times 10^6$), and non-ILC2$^{10}$ ($1 \times 10^6$) were adoptively transferred into islet transplant C57BL/6 mice. In parallel, these three types of ILC2s labeled with CFSE were transfused into islet transplant C57BL/6 mice. All mice were euthanized at day 5 post-islet transplantation. The number of CFSE-labeled ILC2s was quantitated in 4–6 non-overlapping high power fields of islet allograft. For co-transplantation experiments, CD45.2$^+$ ILC2$^{10}$ ($1 \times 10^6$ or $2 \times 10^5$/mouse) were co-transplanted with islets into kidneys of diabetic CD45.2$^+$ or CD45.1$^+$ congenic C57BL/6 mice. The frequency and phenotypes of locally transferred ILC2$^{10}$ were examined at days 5 and 80 post-islet transplantation or at the day when grafts were considered rejected. Graft function was monitored using blood glucose measurements.

### IL-33 and IL-2/IL-2Ab complex administration

IL-2/anti-IL-2 mAb (JES6-1) complexes (IL-2C) were prepared as previously reported (Polhill *et al*, 2012). IL-33 (0.3 μg) alone or IL-33 (0.3 μg) and IL-2C (1 μg/5 μg) was administered intraperitoneally to IL-10-GFP C57BL/6 for five consecutive days. Frequency of ILC2 and ILC2$^{10}$ in kidneys was analyzed by flow cytometry at day 3 post-administration.

### ILC2$^{10}$ induction in vitro

Kidney ILC2s ($2 \times 10^4$/well) isolated from normal IL-10-GFP reporter mice were cultured in round-bottom 96-well plates with RPMI 1640 medium containing IL-7 (20 ng/ml), IL-33 (50 ng/ml),

and IL-2C (20 ng/ml of IL-2; 100 ng/ml of anti-IL-2) for 6 days. Proportion of ILC2[10] was assessed by flow cytometry. In parallel, kidney ILC2s isolated from C57BL/6 mice were cultured with IL-33 and IL-2C for 30 min. The expression of phosphorylated STAT5 was examined by flow cytometry. STAT5 inhibitor (50 μM, CAS 285986-31-4, Sigma) was used to suppress IL-10 production in ILC2s. The secreted cytokine IL-10 in culture supernatants was analyzed via ELISA.

### Cell suspension preparation

Spleen was isolated and digested with collagenase D (1 mg/ml, Roche) and DNase I (100 μg/ml, Roche) for 30 min at 37°C. Kidney was perfused with saline before removal and digested with collagenase IV (1 mg/ml, Sigma) and DNase I (100 μg/ml, Roche) for 40 min at 37°C as previously described (Cao *et al*, 2015). The digested cell suspensions were then passed through a 40-μm cell strainer. Tregs and ILC2s in single-cell suspensions from spleen and kidney were analyzed by flow cytometry.

### Flow cytometry and cell sorting

For FACS analysis of different organ samples, single-cell suspensions were stained with Fc block/anti-CD16/32 (1:200; 2.4G2) and antibodies to CD45.2 (1:100; 104), CD127 (1:100; A7R34), GATA3 (1:40; TWAJ), ST2 (1:50; RMST2-2), as well as antibodies to immune cell lineages (referred as lin): CD3 (1:100; 145-2C11), CD5 (1:200; 53-7.3), TCRβ (1:100; H57-597), TCRγδ (1:100; eBioGL3), CD19 (1:100; 1D3), B220 (1:100; RA3-6B2), CD49b (1:100; DX5), CD11b (1:200; M1/70), CD11c (1:100; N418), FcεRIα (1:100; MAR-1), Gr-1 (1:200; RB6-8C5), and Ter-119 (1:100). Other antibodies used in this study included those against CD4 (1:100; GK1.5), Foxp3 (1:50; FJK-16s), CD25 (1:100; PC61), IL-10 (1:50; JES5-16E3), phospho STAT5 (1:100; SRBCZX), and corresponding isotype controls, all purchased from eBioscience or BioLegend. Cells were analyzed on an LSR Fortessa flow cytometer. For FACS sorting, single-cell suspensions were pregated on hematopoietic cells using anti-CD45.2 antibody, then lineage markers were used to exclude immune cells, and DAPI was used to exclude dead cells. ILC2 [CD45⁺Lin(−)CD127⁺ST2⁺], ILC2[10] [CD45⁺Lin(−)CD127⁺ST2⁺IL-10-GFP], or non-ILC2[10] [CD45⁺Lin(−)CD127⁺ST2⁺IL-10(−)] were sorted using a FACSAria II (BD).

### Quantitative PCR

Total RNA was isolated from allograft by RNeasy Mini Kit (Qiagen) and then reverse-transcribed with First Strand cDNA Synthesis Kit (Fermentas). Real-time PCR was performed on the CFX384 (Bio-Rad) using the SYBR mastermix (Invitrogen). The analysis method was as described before, and the PCR primer sequences are presented in Appendix Table S1.

### Immunohistochemistry and immunofluorescence

Immunohistochemistry of insulin staining was performed as described previously. Briefly, paraffin sections were stained for insulin using polyclonal guinea pig anti-insulin (Dako). The secondary antibody, HRP-conjugated rabbit anti-guinea pig (Dako),

### The paper explained

#### Problem
Pancreatic islet transplantation is a promising treatment option for patients with type 1 diabetes. However, islet graft rejection remains one of the main obstacles to successful transplantation. Clinically applicable strategies for immunomodulation need to be developed to achieve long-term graft tolerance. One attractive therapy to prevent allograft rejection relies on harnessing the potential of regulatory immune cells. Mounting evidence indicates that ILC2s play immune regulatory roles in acute and chronic inflammatory diseases. The current study explores whether ILC2s could suppress allograft rejection in an islet transplantation model.

#### Results
Here, we show that IL-33 treatment significantly prevented islet allograft rejection and improved islet function. This remarkable therapeutic benefit was shown to be mediated by inducing regulatory immune cells including Tregs and ILC2s. A short course of IL-33 treatment induced a sustained increase in ILC2 abundance in islet graft which was associated with IL-33-mediated islet graft protection. Importantly, we further demonstrated that IL-10-producing ILC2s (ILC2[10]), a subset of ILC2, are important inhibitors of islet graft rejection. Co-transplantation of ILC2[10] with islets led to long-term allograft survival, suggesting that ILC2[10] cells are required within the allograft for maximal suppressive effect and graft protection.

#### Impact
Our study offers new insights into the role of IL-33 and ILC2[10] in islet allograft survival. We propose administration of IL-33 and ILC2[10] as an adjunctive therapy to prevent allograft rejection, bringing potential novel therapeutics to the field of transplantation.

was used to detect insulin. Sections were visualized with diaminobenzidine and were counterstained with hematoxylin.

For immunofluorescence staining of ILC2s, frozen sections were stained with polyclonal rabbit anti-mouse CD127 (1:200; Thermofisher), rat anti-mouse ST2 (1:20; RMST2-2), hamster anti-mouse CD3e (1:100; 145-2C11), and polyclonal guinea pig anti-insulin (1:100; Dako) antibodies, and then incubated with the secondary antibodies, AF488 goat anti-rabbit IgG, AF546 goat anti-rat IgG, and AF647 goat anti-hamster IgG and AF405 goat anti-guinea pig. For immunofluorescence staining of Tregs, rabbit monoclonal anti-mouse CD4 (1:50; EPR19514), rat anti-mouse Foxp3 (1:50; FJK-16s) and polyclonal guinea pig anti-insulin (Dako) was used as the primary antibody and AF488 goat anti-rabbit IgG, AF546 goat anti-rat IgG, and AF405 goat anti-guinea pig as the secondary antibodies. Control rabbit, rat, hamster, and guinea pig IgG to primary antibodies were included in staining. The sections were viewed under an Olympus FV1000 (Olympus). The numbers of ILC2s [CD3(−)CD127⁺ST2⁺] and Tregs [CD4⁺Foxp3⁺] were quantitated in 4–6 non-overlapping high power fields of islet allograft. The numbers of ILC2s [CD3(−)CD127⁺ST2⁺] were quantitated in 8–10 non-overlapping high power fields of liver, lung, and kidney.

### Statistics

Statistical tests included unpaired, two-tailed Student's *t*-test using Welch's correction for unequal variances and one-way ANOVA with Tukey's multiple comparison test. Graft survival was analyzed using

the Kaplan–Meier method, and survival curves were compared using the log-rank test. The animals were randomized according to their body weight and blood glucose before starting therapy to avoid any bias and analysis was blinded. Statistical analyses were performed using Prism (Version 7, GraphPad). The exact *P*-values are listed in Appendix Table S2. Results are expressed as the mean ± SEM. A *P* < 0.05 was considered statistically significant.

## Data availability

Additional data are provided in the Appendix and are available online.

**Expanded View** for this article is available online.

### Acknowledgements
We thank the WSLHD and XMU animal facilities for their conscientious care of mice. This work was supported by the National Natural Science Foundation of China grants U1804167 (to Q.H., and Z.N.), 81570624 (to Q.C., Y.W., and Z.N.), 81770721 (to Q.C., and Q.H.), and 81671752 (to W.W., and X.M.); and the National Health & Medical Research Council of Australia grants 1141330 (to Y.W., Q.C., and D.H.) and 1146156 (to Q.C.).

### Author contributions
QC, WW, and DCHH designed and supervised the study. QH, XM, YW, ZN, RW, GL, and MW performed animal experiments and *in vitro* experiments, and analyzed data; QH, XM, RW, YW, ZN, and FY contributed to analysis of islet transplantation model. PR, HW, and WW provided reagents and expertise; QC, WW, HW, YW, and DCHH participated in discussions, provided intellectual input, and wrote the paper.

### Conflict of interest
The authors declare that they have no conflict of interest.

### For more information
Dr. Qi Cao's website: https://www.westmeadinstitute.org.au/research/research-divisions/infection-and-immunity/renal-inflammation-and-immunology-group/overview

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
