## [Review Process File · EMBO Molecular Medicine]

IL-10 producing type 2 innate lymphoid cells prolong islet allograft survival

Qingsong Huang, Xiaoqian Ma, Yiping Wang, Zhiguo Niu, Ruifeng Wang, Fuyan Yang, Menglin Wu, Guining Liang, Pengfei Rong, Hui Wang, David Harris, wei wang, and Qi Cao

DOI: [10.15252/emmm.202012305](https://doi.org/10.15252/emmm.202012305)

Corresponding authors: Qi Cao (qi.cao@sydney.edu.au) , wei wang (cjr.wangwei@vip.163.com)

Review Timeline:

Submission Date:	9th Mar 20
Editorial Decision:	8th Apr 20
Revision Received:	7th Aug 20
Editorial Decision:	10th Sep 20
Revision Received:	15th Sep 20
Accepted:	16th Sep 20

Editor: Zeljko Durdevic

Transaction Report:

8th Apr 2020

Dear Prof. Cao,

Thank you for the submission of your manuscript to EMBO Molecular Medicine. We have now heard back from the two referees who agreed to evaluate your manuscript. As you will see from the reports below, the referees acknowledge the interest of the study. However, they raise some concerns that should be addressed in a major revision of the present manuscript. Addressing the reviewers' concerns in full will be necessary for further considering the manuscript in our journal.

Acceptance of the manuscript will entail a second round of review. Please note that EMBO Molecular Medicine encourages a single round of revision only and therefore, acceptance or rejection of the manuscript will depend on the completeness of your responses included in the next, final version of the manuscript. For this reason, and to save you from any frustrations in the end, I would strongly advise against returning an incomplete revision.

We realize that the current situation is exceptional on the account of the COVID-19/SARS-CoV-2 pandemic. Therefore, please let us know if you need more than three months to revise the manuscript.

I look forward to receiving your revised manuscript.

***** Reviewer's comments *****

Referee #1 (Remarks for Author):

Overall this is an innovative paper which offers a novel way of prolonging the survival of transplanted islets. The experiments were performed carefully, however some additional data is required to be completely convincing:

1. IL-33 treatment was shown to have glucose-lowering effects on its own that are both insulin-dependent and insulin-independent (Molofsky et al., 2013; Dalmás et al., 2017). Considering that islet transplantation is performed after IL-33 treatment, authors should show the survival curve and glycemia of STZ-induced diabetic mice treated with IL-33 (but without islet graft). These data are required to control that prolonged islet allograft survival is not also due to a better glycemia status at the time of islet graft.
2. In Fig 1D, GTT was done only with 3-5 mice although 12 mice survived the 80 day time point (Fig 1 b). How were these few mice selected for the GTT?
3. The sustained ILC2s increase in the islets graft, long after exogenous IL-33 treatment, is of great interest. IL-33 have been shown to be locally produced in mouse islets, and especially in Balb/c mouse islets that are grafted here (Dalmás et al., 2017). Did the authors check for sustained increased expression of IL-33 (or any other ILC2-promoting factors) in the islet graft over time? This could partly explain why ILC2s stay inside the graft so long.
4. Fig 6A shows in vitro the reduced production of IL10 upon knockdown CRISPR ko of IL10 in ILC210. Can this reduction only be detected in isolated cells in vitro or also in whole organs, does it affects IL10 plasma levels?
5. Figure 7 Do ILC2s from the local transplantation with the graft migrate in other organs? Does islet rejection occur because of the ILC2 death in the graft? Authors should show the staining of ILC2 in

islet grafts of islet transplant mice over time until rejection.

6. Please describe how the islets were isolated. Also, the number of islets isolated per mice seems unusually high, please control
7. PC61 (anti-CD25 antibody) is not described in methods section, provider?
8. Mice: sex of mice is missing.
9. DEREK and IL-10-GFP transgenic mice should be better described, source, origin reference to production of mouse strains.
10. Description of islet transplantation is poor and also no reference to the details of the protocol is provided.
11. "Neutralizing anti-IL-10 antibodies (10 µg/ml Biolegend)", specify antibody used.
12. Provide rationale for IL-2/anti-IL2 mAB.
13. Description/Provider of ELISA assay for IL10 is missing
14. The description of the CRISPR-Cas9 transfection is very superficial.

Referee #2 (Remarks for Author):

In the manuscript entitled "IL-10 producing type 2 innate lymphoid cells induced by IL-33 prolong islet allograft survival," the authors present data that show that systemic IL-33 treatment of diabetic mice that receive islet allografts, prolongs the survival of the transplanted islets by increasing the levels of Tregs and especially IL10 secreting, ILC2 cells. The data are intriguing, but the manuscript would be strengthened with the follow data:

- 1) Whether, after systemic IL-33 treatment, ILC2 cells localize to other non-lymphoid organs (e.g., liver, heart, the kidney that did not receive islets) versus localizing to where the islet allografts are found and where an immune response is under way.
- 2) The relative number of infiltrating CD4+ versus CD8+ T cells in vehicle versus IL-33 treated animals to provide insight into the mechanism of rejection/acceptance in the experimental and control groups.
- 3) A control group in Fig 4 of islet + IL-33/PC61 without DT. Also, the authors show that Treg levels peak at day 7 in the kidney but ILC2 levels remain high up to day 30 in the kidney and up to day 80 within the islet allografts. Have the authors treated the recipients with PC61 at day 30 and/or day 80? In addition, have they treated recipients with DT at those times? It could be that Tregs don't play as great a role in the later time points in IL-33 treated recipients and these additional studies could help explain the results in Figure 4D where 40% graft survival is observed after DT treatment.
- 4) In Figure 6F, the authors show suppressive activity by ILC210 cells in vitro. Have the authors performed similar experiment using ILC210 -IL10 cells? If so, are ILC210 -IL10 cells as suppressive and if not, does the addition of IL10 result in the recovery of their suppressive activity?
- 5) Regarding Figure 7, does injection of ILC210 -IL10 cells injected locally result in decreased survival of the islet allografts?

Referee #1 (Remarks for Author):

Overall this is an innovative paper which offers a novel way of prolonging the survival of transplanted islets. The experiments were performed carefully, however some additional data is required to be completely convincing:

1. IL-33 treatment was shown to have glucose-lowering effects on its own that are both insulin-dependent and insulin-independent (Molofsky et al., 2013; Dalmas et al., 2017). Considering that islet transplantation is performed after IL-33 treatment, authors should show the survival curve and glycemia of STZ-induced diabetic mice treated with IL-33 (but without islet graft). These data are required to control that prolonged islet allograft survival is not also due to a better glycemia status at the time of islet graft.

Thanks for this valuable suggestion, following which we examined the survival rate and glycemia of STZ-induced diabetic mice treated with IL-33. The additional experiments showed that IL-33 treatment (starting from day 6 after STZ injection) in diabetic mice without islet transplantation improved the fasting and non-fasting glycemia at day 15 and 18 post-STZ injection, but not at other time points. IL-33 treatment did not enhance survival of STZ-induced diabetic mice within 30 days (new Figure S1). These results indicate that short-term IL-33 treatment only temporarily improved hyperglycaemia, which might contribute to prolonged islet allograft survival. However, we further demonstrated that Tregs and ILC2s played critical roles in IL-33-mediated islet graft protection (Figures 4, 6 and 7).

2. In Fig 1D, GTT was done only with 3-5 mice although 12 mice survived the 80 day time point (Fig 1 b). How were these few mice selected for the GTT?

We conducted two independent animal experiments in Figure 1, but only showed the data from one set of animal experiments in Figure 1D and E. We now added all data in Figure 1.

3. The sustained ILC2s increase in the islets graft, long after exogenous IL-33 treatment, is of great interest. IL-33 have been shown to be locally produced in mouse islets, and especially in Balb/c mouse islets that are grafted here (Dalmas et al., 2017). Did the authors check for sustained increased expression of IL-33 (or any other ILC2-promoting factors) in the islet graft over time ? This could partly explain why ILC2s stay inside the graft so long.

According to the reviewer's suggestion, the expression of ILC2-promoting factors, including IL-25, IL-33 and thymic stromal lymphopoietin (TSLP) were examined in islet graft tissue at day 7, 30 and 80 post-islet transplantation. We observed a consistent increase of IL-33, but not IL-25 or TSLP, in islet graft tissue of mice treated with IL-33, which could partly explain why ILC2s were found within the graft for so long. However, the reason for the consistent increase of IL-33 in islet graft needs future investigation.

4. Fig 6A shows in vitro the reduced production of IL10 upon knockdown CRISPR ko of IL10 in ILC210. Can this reduction only be detected in isolated cells in vitro or also in whole organs, does it affects IL10 plasma levels?

Additional experiments were performed to examine expression of IL-10 in islet graft and in plasma. The expression of IL-10 in islet graft tissue was significantly increased in mice transfused with ILC2¹⁰, but reduced in mice transfused with IL-10-deleted ILC2¹⁰ (Figure 6D). There was no significant change of IL-10 levels in plasma in mice transfused with ILC2¹⁰, IL-10-deleted ILC2¹⁰ or non-ILC2¹⁰. These results indicate that transfused ILC2¹⁰ only affect IL-10 expression in local islet graft tissue, but not systemic IL-10 levels in plasma.

5. Figure 7 Do ILC2s from the local transplantation with the graft migrate in other organs? Does islet rejection occur because of the ILC2 death in the graft? Authors should show the staining of ILC2 in islet grafts of islet transplant mice over time until rejection.

Thanks for another valuable suggestion. Additional experiments using CD45.2 and CD45.1 mice were performed to examine the number and phenotype of locally transferred ILC2¹⁰ in islet graft over time (new Figure 7C-F). There was no difference in the number of CD45.2+ ILC2¹⁰ in rejected islet graft and in accepted islet graft (survival for 80 days post-islet transplantation) (Figure 7D and E), suggesting that the occurrence of islet rejection is not because the number of ILC2 reduces in the graft over time. We did not detect CD45.2+ ILC2¹⁰ in other organs, including kidney, liver and lung, by flow cytometry in recipient mice with local ILC2¹⁰ transplantation. Furthermore, we found phenotypic changes of ILC2¹⁰ in rejected islet graft, especially reduced expression of IL-10 (Figure 7F and Figure S5), which could possibly explain why locally transferred ILC2¹⁰ did not lead to long-term islet graft survival in 5 out of 9 islet transplanted mice.

6. Please describe how the islets were isolated. Also, the number of islets isolated per mice seems unusually high, please control

We have added the protocol of islets isolation in the revised manuscript. Pancreatic islets were separated from the pancreata of donor (BALB/c) mice at a ratio of four pancreata per recipient. The “2000 IEQ/mouse” means “2000 IEQ per recipient mouse”. We have made correction in method section.

7. PC61 (anti-CD25antibody) is not described in methods section, provider?

We have added the source for anti-CD25 antibody, PC61.

8. Mice: sex of mice is missing.

Male mice were used in all animal experiments, which has been described on page 18 in the Methods section.

9. DEREK and IL-10-GFP transgenic mice should be better described, source, origin reference to production of mouse strains.

We have added the information for DEREK and IL-10-GFP transgenic mice.

10. Description of islet transplantation is poor and also no reference to the details of the protocol is provided.

The detailed protocol and reference have been added in the Methods section.

11. "Neutralizing anti-IL-10 antibodies (10 µg/ml Biolegend)", specify antibody used.

The clone number of anti-IL-10 antibody (JES5-16E3) has been added in the Methods section.

12. Provide rationale for IL-2/anti-IL2 mAB.

The rationale for using IL-2/anti-IL2 mAB is added in the revised manuscript (page 10).

13. Description/Provider of ELISA assay for IL10 is missing

The description and provider of ELISA assay have been added in the Methods section.

14. The description of the CRISPR-Cas9 transfection is very superficial.

A detailed description of the CRISPR-Cas9 transfection has been added in the Methods section.

Referee #2 (Remarks for Author):

In the manuscript entitled "IL-10 producing type 2 innate lymphoid cells induced by IL-33 prolong islet allograft survival," the authors present data that show that systemic IL-33 treatment of diabetic mice that receive islet allografts, prolongs the survival of the transplanted islets by increasing the levels of Tregs and especially IL10 secreting, ILC2 cells. The data are intriguing, but the manuscript would be strengthened with the follow data:

1) Whether, after systemic IL-33 treatment, ILC2 cells localize to other non-lymphoid organs (e.g., liver, heart, the kidney that did not receive islets) versus localizing to where the islet allografts are found and where an immune response is under way.

Thanks for your very helpful suggestions. We have previously shown that systemic IL-33 treatment induced ILC2 expansion in non-lymphoid organs, such as kidney, liver and lung.¹ Considering that islet transplantation is performed after IL-33 treatment, we proposed that ILC2s will migrate into islet graft after islet transplantation. We found that a greater amount of ILC2s were found in islet graft than in kidney and liver (Figure S3), which indicates that ILC2s tend to migrate to islet graft undergoing immune response.

2) The relative number of infiltrating CD4+ versus CD8+ T cells in vehicle versus IL-33 treated animals to provide insight into the mechanism of rejection/acceptance in the experimental and control groups.

Additional experiments were performed to examine the ratio of infiltrating CD4+ versus CD8+ T cells in mice treated with vehicle or IL-33. We observed a significant increase of ratio of CD4 T cells/CD8 T cells in islet graft of mice treated with IL-33 (Figure. S2), suggesting that IL-33 treatment may prevent islet graft rejection through modulating local CD4 and CD8 T cell responses.

3) A control group in Fig 4 of islet + IL-33/PC61 without DT. Also, the authors show that Treg levels peak at day 7 in the kidney but ILC2 levels remain high up to day 30 in the kidney and up to day 80 within the islet allografts. Have the authors treated the recipients with PC61 at day 30 and/or day 80? In addition, have they treated recipients with DT at those times? It could be that Tregs don't play as great a role in the later time points in IL-33 treated recipients and these additional studies could help explain the results in Figure 4D where 40% graft survival is observed after DT treatment.

Following the reviewer's suggestion, a control group of islet + IL-33/PC61 without DT treatment was included in new Figure 4. Administration of anti-CD25 antibody (PC61) successfully depleted both Tregs and ILC2s in vivo as CD25 is highly expressed on both Tregs and ILC2s. Tregs and ILC2 depletion by PC61 completely abolished the protective effects of IL-33 on islet transplantation, indicating both Tregs and ILC2s play critical roles in IL-33-mediated islet graft protection. However, we have not treated the recipients with PC61 at day 30 and/or day 80 because the recipient mice that received PC61 treatment prior to islet transplantation rejected their graft within 30 days. We have not performed later stage depletion of Tregs (DT treatment at day 30 or 80) because IL-33 induced Tregs levels that peaked at day 7 and were down to normal at day 14 (Figure 2 and 3). Therefore, DT treatment at day 30 or 80 could not be used to demonstrate the protective effects of Tregs which were induced by IL-33. Additional ILC2 depletion in Treg-depleted DEREK mice completely abolished the protective effects of IL-33 on islet transplantation (Figure 4), suggesting that ILC2, but not Tregs, play an important role in the later time points in IL-33 treated recipients.

4) In Figure 6F, the authors show suppressive activity by ILC2¹⁰ cells in vitro. Have the authors performed similar experiment using ILC2¹⁰ -IL10 cells? If so, are ILC2¹⁰ -IL10 cells as suppressive and if not, does the addition of IL10 result in the recovery of their suppressive activity?

Following the reviewer's suggestion, we performed similar experiments using IL-10-deleted ILC2¹⁰, shown in Figure 6G. ILC2¹⁰ effectively suppressed allogeneic splenocyte induced CD4 T cell proliferation in a dose dependent manner, and genetic ablation of IL-10 diminished the suppressive role of ILC2¹⁰ on CD4 T cell proliferation (new Figure 6F and 6G), indicating that IL-10 is an important mediator in ILC2¹⁰-mediated immune suppression. Regarding the suggestion of adding back IL-10 into cell culture system, this could not be used to demonstrate the recovery of suppressive effect of ILC2¹⁰, as IL-10 alone has immunosuppressive effects.

5) Regarding Figure 7, does injection of ILC2¹⁰ -IL10 cells injected locally result in decreased survival of the islet allografts?

Additional experiments showed that local transfer IL-10-deleted ILC2¹⁰ failed to prolong islet graft survival in comparison with ILC2¹⁰ (new Figure 7). These data further confirmed that ILC2¹⁰ prolonged islet graft survival in an IL-10-dependent manner.

Reference

1. Cao Q, Wang Y, Niu Z, *et al.* Potentiating Tissue-Resident Type 2 Innate Lymphoid Cells by IL-33 to Prevent Renal Ischemia-Reperfusion Injury. *J Am Soc Nephrol* 2018; **29**: 961-976.

10th Sep 2020

Dear Prof. Cao,

Thank you for the submission of your revised manuscript to EMBO Molecular Medicine. We have now received the enclosed reports from the referees that were asked to re-assess it. As you will see the reviewers are now globally supportive and I am pleased to inform you that we will be able to accept your manuscript pending the following final amendments:

***** Reviewer's comments *****

Referee #1 (Remarks for Author):

The authors made a great effort and addressed well the critics

Responses to your comments:

Referee #1 (Remarks for Author):

The authors made a great effort and addressed well the critics

We thank the reviewer's generous comments.

The authors performed the requested changes.

Corresponding Author Name: QI CAO

Manuscript Number: EMM-2020-12305